# <sup>177</sup>Lu-PSMA-617 in Metastatic Castration-Resistant Prostate Cancer: A Review of the Evidence and Implications for Canadian Clinical Practice

Kim N. Chi [1,*], Steven M. Yip [2], Glenn Bauman [3], Stephan Probst [4], Urban Emmenegger [5], Christian K. Kollmannsberger [1], Patrick Martineau [6], Tamim Niazi [7], Frédéric Pouliot [8,9], Ricardo Rendon [10], Sebastien J. Hotte [11], David T. Laidley [12] and Fred Saad [13,14]

1   Department of Medical Oncology, BC Cancer—Vancouver, University of British Columbia,
    Vancouver, BC V5Z 1M9, Canada; ckollmannsberger@bccancer.bc.ca
2   Department of Oncology, Tom Baker Cancer Centre and Cumming School of Medicine, University of Calgary,
    Calgary, AB T2N 4N1, Canada; steven.yip@albertahealthservices.ca
3   London Regional Cancer Program, Department of Oncology, Western University,
    London, ON N6A 5W9, Canada; glenn.bauman@lhsc.on.ca
4   Department of Nuclear Medicine, Jewish General Hospital, McGill University,
    Montreal, QC H3A 0G4, Canada
5   Department of Medicine, Odette Cancer Centre, University of Toronto, Toronto, ON M5S 1A8, Canada;
    uemmengg@sri.utoronto.ca
6   Department of Radiology, BC Cancer—Vancouver, University of British Columbia,
    Vancouver, BC V5Z 1M9, Canada; patrick.martineau@bccancer.bc.ca
7   Department of Radiation Oncology, Jewish General Hospital, McGill University,
    Montréal, QC H3T 1E2, Canada; mohammad.tamim.niazi.med@ssss.gouv.qc.ca
8   Department of Urology, Centre Hospitalier Universitaire de Québec, Université Laval,
    Québec, QC G1V 0A6, Canada; frederic.pouliot@crchudequebec.ulaval.ca
9   Department of Surgery, Université Laval, Québec, QC G1V 0A6, Canada
10  Department of Urology, Queen Elizabeth II Health Sciences Centre, Dalhousie University,
    Halifax, NS B3H 4R2, Canada
11  Department of Oncology, Juravinski Cancer Centre, McMaster University, Hamilton, ON L8S 4L8, Canada;
    hotte@hhsc.ca
12  Department of Medical Imaging-Nuclear Medicine, London Health Sciences Centre, Western University,
    London, ON N6A 3K7, Canada
13  Division of Urology, Centre Hospitalier de l'Université de Montréal, Université de Montréal,
    Montréal, QC H2X 0A9, Canada; fred.saad@umontreal.ca
14  Department of Surgery, Université de Montréal, Montréal, QC H2X 0A9, Canada
*   Correspondence: kchi@bccancer.bc.ca

**Abstract:** Prostate-specific membrane antigen (PSMA) is highly expressed in prostate cancer and a therapeutic target. Lutetium-177 (<sup>177</sup>Lu)-PSMA-617 is the first radioligand therapy to be approved in Canada for use in patients with metastatic castration-resistant prostate cancer (mCRPC). As this treatment represents a new therapeutic class, guidance regarding how to integrate it into clinical practice is needed. This article aims to review the evidence from prospective phase 2 and 3 clinical trials and meta-analyses of observational studies on the use of <sup>177</sup>Lu-PSMA-617 in prostate cancer and discuss how Canadian clinicians might best apply these data in practice. The selection of appropriate patients, the practicalities of treatment administration, including necessary facilities for treatment procedures, the assessment of treatment response, and the management of adverse events are considered. Survival benefits were observed in clinical trials of <sup>177</sup>Lu-PSMA-617 in patients with progressive, PSMA-positive mCRPC who were pretreated with androgen receptor pathway inhibitors and taxanes, as well as in taxane-naïve patients. However, the results of ongoing trials are awaited to clarify questions regarding the optimal sequencing of <sup>177</sup>Lu-PSMA-617 with other therapies, as well as the implications of predictive biomarkers, personalized dosimetry, and combinations with other therapies.

**Keywords:** prostate cancer; mCRPC; radioligand therapy; $^{177}$Lu-PSMA-617; $^{177}$Lu vipivotide tetraxetan

## 1. Introduction

Prostate cancer accounts for approximately 10% of cancer mortality among males in Canada [1]. Historically, nearly all of these deaths were due to metastatic disease, which in 2011–2019 had a five-year relative survival rate of 34%, as compared to almost 100% for localized or regional disease [2]. The results of recent clinical trials suggest that median overall survival (OS) for patients with previously untreated metastatic castration-resistant prostate cancer (mCRPC) ranges from 31 to 41 months [3–6], with real-world data revealing a slightly shorter median OS of 21 months [7]. Research into therapeutics leveraging novel targets such as prostate-specific membrane antigen (PSMA) has aimed to improve survival in patients with metastatic prostate cancer [8–10]. PSMA is a transmembrane enzyme that has low levels of expression in normal prostate, kidney, and small intestine tissue, as well as salivary and lachrymal glands, but is overexpressed by 100- to 1000-fold in over 90% of metastatic prostate cancers, with particularly elevated levels in mCRPC [8–10]. Various strategies targeting PSMA in prostate cancer have been investigated, including monoclonal antibodies and small-molecule radioligand therapy (RLT) [9,10]. Of these, the only PSMA-directed therapeutic currently approved in Canada is lutetium-177 ($^{177}$Lu)-PSMA-617, also known as $^{177}$Lu vipivotide tetraxetan, although alternative RLTs utilizing different PSMA-binding molecules and/or radionuclides are being evaluated in clinical trials [11,12].

$^{177}$Lu-PSMA-617 is a small-molecule RLT consisting of the radionuclide $^{177}$Lu linked to a PSMA-binding ligand [12]. The radionuclide $^{177}$Lu has a half-life of 6.6 days and emits primarily beta rays, which have an average range of 0.23 mm in soft tissue [13]. The PSMA-binding ligand, PSMA-617, is a chemically modified PSMA inhibitor demonstrated to have high inhibition potency and efficient internalization into PSMA-positive cells [14]. Binding of $^{177}$Lu-PSMA-617 to PSMA-expressing cells delivers radiation in a target-specific manner, leading to cell death of PSMA-positive cells, as well as of surrounding cells due to the cross-fire effect [12,15]. Following promising preclinical results with radiolabeled PSMA-617 developed at the German Cancer Research Center in Heidelberg, Germany [14], $^{177}$Lu-PSMA-617 was successfully used on a compassionate basis in patients with metastatic prostate cancer treated at German centres [16–21]. This paved the way for formal phase 1 dose-escalation trials in mCRPC to determine the recommended phase 2 dose [22,23], and subsequently for the phase 3 VISION trial [24], which allowed regulatory approval across numerous jurisdictions [12,25,26]. More recently, the results of the phase 3 PSMAfore trial evaluating $^{177}$Lu-PSMA-617 earlier in the mCRPC treatment course have been reported [27].

In Canada, $^{177}$Lu-PSMA-617 administered for up to six cycles was approved by Health Canada in 2022 for the treatment of adult patients with PSMA-positive mCRPC who have received at least one androgen receptor pathway inhibitor (ARPI) and at least one taxane-based chemotherapy regimen [12]. The treatment subsequently received a recommendation in the 2022 Canadian Urological Association–Canadian Uro-Oncology Group guideline [28]. The Canadian Agency for Drugs and Technologies in Health also recommended it be reimbursed by public drug plans, though with the caveat that the suggested pricing be reduced as list pricing would result in additional costs of $122,489 per patient [29]. However, as the first RLT to be approved for use in metastatic prostate cancer, there are practical challenges relating to integrating $^{177}$Lu-PSMA-617 into clinical practice. This article aims to review the existing evidence and to discuss how Canadian clinicians might best apply these data for their patients.

## 2. Materials and Methods

Prospective phase 2 and 3 clinical trials and meta-analyses of observational studies on the use of $^{177}$Lu-PSMA-617 in prostate cancer were identified through a search of

the English language literature and recent major congress abstracts. Databases searched included PubMed and Google Scholar, which were used to identify publications from January 2013 to August 2023, as well as the repositories of abstracts presented at the American Society of Clinical Oncology (ASCO), ASCO Genitourinary Cancers (ASCO-GU), European Society for Medical Oncology (ESMO), and Society of Nuclear Medicine and Molecular Imaging (SNMMI) congresses from 2020 to 2023 (Appendix A). Abstracts were evaluated to identify potentially relevant data sources for full review.

## 3. Results

### 3.1. Results of Literature Review

The searches of the Google Scholar and PubMed databases identified 357 and 65 articles, respectively, while 8, 7, and 410 abstracts were identified from searches of the ASCO/ASCO-GU, ESMO, and SNMMI congress abstracts, respectively. After removal of duplicates, a total of 668 records were reviewed, 649 of which were removed after abstract review. A total of 17 publications and 9 congress abstracts/posters were reviewed (Figure 1). Five additional sources not meeting the original search criteria were also proposed by the authors.

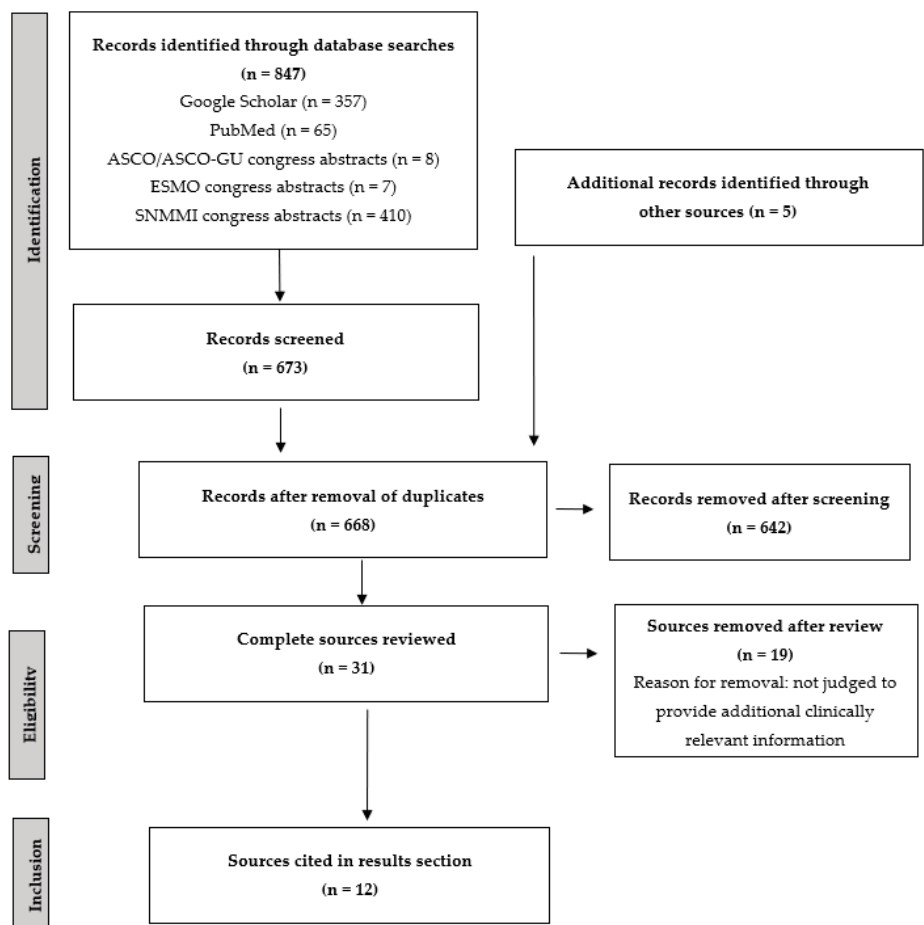

**Figure 1.** PRISMA flow diagram of literature review. ASCO, American Society of Clinical Oncology; ASCO-GU, American Society of Clinical Oncology Genitourinary Cancers; ESMO, European Society for Medical Oncology; PRISMA, Preferred Reporting Items for Systematic Reviews and Meta-Analyses; SNMMI, Society of Nuclear Medicine and Molecular Imaging.

### 3.2. Patients Treated with $^{177}$Lu-PSMA-617

More than 850 patients with progressive mCRPC have been treated with $^{177}$Lu-PSMA-617 in the context of three randomized phase 2 and 3 clinical trials [23,24,27]. While the majority of patients in these trials had an Eastern Cooperative Oncology Group (ECOG) performance

status (PS) of 0 or 1 and the median age ranged from 70 to 72 years, there was considerable variation between the trials in other disease characteristics (Table 1). These differences were partly due to differing inclusion criteria. For instance, median prostate-specific antigen (PSA) levels in the ARPI-treated, taxane-naïve patients in the PSMAfore study were <20 μg/L [27], in contrast to median PSA levels of 75–110 μg/L in the TheraP and VISION trials' taxane-exposed patients [23,24]. The definition of PSMA-positivity criteria also varied across trials, with the TheraP trial mandating both PSMA-based and fluorodeoxyglucose (FDG)-based positron emission tomography (PET) scans [23], as compared to the VISION and PSMAfore trials, which used PSMA PET and contrast computed tomography (CT) scan correlation to determine study eligibility [24,27].

**Table 1.** Baseline patient characteristics in selected randomized phase 2 and 3 studies of $^{177}$Lu-PSMA-617 in progressive PSMA-positive mCRPC.

| | TheraP [23] | | VISION [24] | | PSMAfore [27] | |
|---|---|---|---|---|---|---|
| **Study type** | Phase 2 | | Phase 3 | | Phase 3 | |
| **PSMA PET eligibility criteria** | $^{68}$Ga-PSMA-11, SUV$_{max}$ ≥ 20 at ≥1 disease site and >10 at all other metastatic disease sites | | $^{68}$Ga-PSMA-11 uptake greater than liver parenchyma at ≥1 disease site and no PSMA-negative metastatic lesions | | $^{68}$Ga-PSMA-11 uptake greater than liver parenchyma at ≥1 disease site and no PSMA-negative metastatic lesions | |
| **FDG PET eligibility criteria** | No sites with discordant FDG-positive/ PSMA-negative lesions | | N/A | | N/A | |
| **Study arms** | LuPSMA | Cabazitaxel | LuPSMA | SOC | LuPSMA | ARPI change |
| **Patients, n** | 99 | 101 | 551 | 280 | 234 | 234 |
| **Median age, years** | 72.1 | 71.8 | 70.0 | 71.5 | 71 | 72 |
| **ECOG PS 0 or 1, %** | 96 | 96 | 92.6 | 92.1 | 99.1 | 97.9 |
| **Median PSA level, μg/L** | 93.5 | 110 | 77.5 | 74.6 | 18.4 | 14.9 |
| **Median ALP level, IU/L** | 111 | 130 | 105.0 | 94.5 | 100.0 | 103.5 |
| **Disease sites, %** | | | | | | |
| Bone | 90.9 | 89.1 | 91.5 | 91.4 | 87.6 | 86.8 |
| Liver | 7.11 [1] | 12.91 [1] | 11.4 | 13.6 | 5.6 | 3.0 |
| Lymph node | 52.5 | 46.5 | 49.7 | 50.4 | 32.5 | 31.6 |
| **Previous treatments, %** | | | | | | |
| ARPI | 92 | 90 | 100 | 100 | 100 | 100 |
| Cabazitaxel | 0 | 0 | 37.9 | 38.2 | 0 | 0 |
| Docetaxel | 100 | 100 | 96.9 | 97.5 | 0 | 0 |

[1] Includes all visceral disease sites (lung, liver, and other), not just liver. $^{68}$Ga, gallium-68; $^{177}$Lu, lutetium-177; ALP, alkaline phosphatase; ARPI, androgen receptor pathway inhibitor; ECOG, Eastern Cooperative Oncology Group; FDG, fluorodeoxyglucose; LuPSMA, $^{177}$Lu-PSMA-617; mCRPC; metastatic castration-resistant prostate cancer; N/A, not applicable; PET, positron emission tomography; PS, performance status; PSA, prostate-specific antigen; PSMA, prostate-specific membrane antigen; SOC, standard of care; SUV$_{max}$, maximum standard uptake value.

Outside of clinical trials, the use of $^{177}$Lu-PSMA-617 has been reported in more than 2500 patients with progressive mCRPC in real-world settings [30]. The reported median age of patients in one systematic review of observational studies ranged from 65 to 72 years [31]. In contrast to clinical trials, these patients had higher median PSA levels ranging from 59 to 1000 μg/L, along with a greater proportion of liver metastases (18%) [32]. Approximately 70% of the patients included in such observational studies had previously been treated with a taxane [32].

*3.3. Survival Outcomes with $^{177}$Lu-PSMA-617*

In the VISION trial, which enrolled patients with mCRPC who had received at least one line of ARPI treatment and one line of chemotherapy, $^{177}$Lu-PSMA-617 improved imaging-based progression-free survival (PFS) by 60% (hazard ratio [HR] = 0.40; 95% confidence interval [CI] 0.29–0.57; $p < 0.001$) and OS by 38% (HR = 0.62; 95% CI 0.52–0.74; $p < 0.001$) vs. trial-permitted best standard of care (SOC) (Table 2) [24]. The median imaging-based PFS was 8.7 months and median OS was 15.3 months for the $^{177}$Lu-PSMA-617 arm, vs. 3.4 months and 11.3 months, respectively, for the SOC arm. In the TheraP trial, which enrolled patients with mCRPC who had received prior docetaxel chemotherapy,

treatment with $^{177}$Lu-PSMA-617 also significantly increased PFS (defined as the interval from randomization to first evidence of PSA progression) vs. cabazitaxel (HR = 0.63; 95% CI 0.46–0.86; *p* = 0.0028) [23]. In PSMAfore, which enrolled patients who were chemotherapy-naïve, at the time of the second interim analysis median imaging-based PFS was 6.4 months longer in the $^{177}$Lu-PSMA-617 arm vs. the ARPI change arm (HR = 0.43; 95% CI 0.33–0.54; *p* < 0.0001), while there was no significant difference in median OS, which was 19.25 vs. 19.71 months in the $^{177}$Lu-PSMA-617 and ARPI change arms, respectively (HR = 1.18; 95% CI 0.83–1.64) [27]. It is important to note that a high crossover rate occurred, with 84.2% of patients who progressed in the ARPI change arm subsequently receiving $^{177}$Lu-PSMA-617, thus likely diminishing the between-arm differences [27].

**Table 2.** Survival and quality of life outcomes in selected randomized phase 2 and 3 studies of $^{177}$Lu-PSMA-617 in progressive PSMA-positive mCRPC.

| | TheraP [23] | | VISION [24,33] | | PSMAfore [27] | |
|---|---|---|---|---|---|---|
| **Study arms** | **LuPSMA** | **Cabazitaxel** | **LuPSMA** | **SOC** | **LuPSMA** | **ARPI Change** |
| **Patients, n** | 99 | 101 | 551 | 280 | 234 | 234 |
| **Median imaging-based PFS, months** | NR | NR | 8.7 | 3.4 | 12.0 | 5.6 |
| HR (95% CI) | | 0.63 (10.46–0.86) | | 0.40 (0.29–0.57) | | 0.41 (0.33–0.54) |
| *p* value | | 0.0028 | | <0.001 | | <0.0001 |
| **Median OS, months** | NR | NR | 15.3 | 11.3 | 19.2 | 19.7 |
| HR (95% CI) | | | | 0.62 (0.52–0.75) | | 1.16 (0.83–1.64) |
| *p* value | | NR | | <0.001 | | NR |
| **Median time to HRQOL worsening, months** [1] | NR | NR | 14.3 | 2.9 | 7.5 | 4.3 |
| HR (95% CI) | | | | 0.45 (0.33–0.60) | | 0.59 (0.47–0.72) |
| *p* value | | NR | | <0.001 | | NR |
| **Median time to pain worsening, months** [2] | NR | NR | 1.0 | 0.5 | 5.0 | 3.7 |
| HR (95% CI) | | | | 0.65 (0.54–0.78) | | 0.69 (0.56–0.85) |
| *p* value | | NR | | <0.001 | | NR |

[1] As measured by FACT-P score; [2] As measured on BPI-SF scale. $^{177}$Lu, lutetium-177; LuPSMA, $^{177}$Lu-PSMA-617; ARPI, androgen receptor pathway inhibitor; BPI-SF, Brief Pain Inventory-Short Form; CI, confidence interval; FACT-P, Functional Assessment of Cancer Therapy-Prostate; HR, hazard ratio; HRQOL, health-related quality of life; mCRPC; metastatic castration-resistant prostate cancer; NR, not reported; OS, overall survival; PFS, progression-free survival; PSMA, prostate-specific membrane antigen; SOC, standard of care.

A meta-analysis of observational studies of $^{177}$Lu-PSMA RLT found that median OS in the real-world setting was 16 months [31]. This meta-analysis concluded that survival was longer in chemotherapy-naïve vs. chemotherapy-resistant patients, those with an ECOG PS of 0 vs. 1–2, those with only lymph node metastases vs. those with bone, lung, or liver metastases, those with normal vs. elevated serum alkaline phosphatase (ALP), those with higher vs. lower average standard uptake values ($SUV_{average}$) and minimal SUV ($SUV_{min}$), those who received an intensified vs. conventional schedule of RLT, and those who had a PSA decline of at least 50% [31]. Other meta-analyses confirmed the negative impact of visceral metastases and prior taxane-based chemotherapy on OS following $^{177}$Lu-PSMA RLT [32,34].

Similar results were seen in phase 2 and 3 clinical trials. For instance, while no OS benefit has yet been shown in taxane-naïve patients treated with $^{177}$Lu-PSMA-617 vs. ARPI change, possibly due to high levels of crossover in the phase 3 PSMAfore trial [27], subgroup analyses from the VISION trial suggested that both imaging-based PFS and OS benefits were potentially greater in patients who had previously been treated with one vs. two or more taxanes [35]. The same analyses also suggested greater survival benefits in patients who had been treated with at least two vs. only one ARPI, as well as those not on concurrent ARPI vs. on concurrent ARPI [35]. Additionally, the results of both TheraP and VISION suggested patients with higher whole-body tumour mean SUV on PSMA PET

(SUV$_{mean}$ ≥ 10) were more likely to derive imaging-based PFS benefit from treatment with [177]Lu-PSMA-617 than those with lower SUV$_{mean}$, although all subgroups benefitted [36,37].

A multicentre retrospective study analyzed data from 176 patients treated with [177]Lu-PSMA RLT in order to incorporate these predictive markers into nomograms, which were then validated in another cohort of 74 patients [38]. Factors in the OS nomogram included time since diagnosis, use of previous chemotherapy, tumour SUV, and presence of pelvic nodal, bone, and liver metastases [38].

### 3.4. Quality of Life with [177]Lu-PSMA-617

Quality of life analyses of the VISION trial found that treatment with [177]Lu-PSMA-617 in addition to SOC delayed time to worsening vs. SOC alone in terms of measures of health-related quality of life (HRQOL) and pain, such as the Functional Assessment of Cancer Therapy-Prostate (FACT-P) score and Brief Pain Inventory-Short Form (BPI-SF) pain intensity score ($p < 0.001$ for all comparisons, Table 2) [33]. Quality of life analyses of the PSMAfore study found similar benefits in delaying time to worsening HRQOL and pain in the [177]Lu-PSMA-617 vs. ARPI change arms [27].

### 3.5. Adverse Events Associated with [177]Lu-PSMA-617

The adverse events of any grade that were most commonly increased in the [177]Lu-PSMA-617 + SOC arm vs. the SOC alone arm in the VISION trial included dry mouth, fatigue, nausea, anemia, and diarrhea, while the most commonly increased grade ≥3 adverse events were anemia, thrombocytopenia, and lymphopenia, which were generally infrequent (Figure 2) [24]. In this study, treatment-emergent adverse events occurred with similar frequency during cycles 1–5, which had median durations of 6 weeks each, with more adverse events being observed during cycle 6, which had a median duration of 26 weeks since the period of observation continued beyond week 6, reflecting an ascertainment bias due to the longer observation period [39]. Increases of ≥10% in the incidence of dry mouth with [177]Lu-PSMA-617 vs. control were also observed in the other randomized clinical trials, TheraP and PSMAfore, which employed cabazitaxel and ARPI change as controls, respectively [23,27]. Increases of ≥10% in the incidence of thrombocytopenia was also noted in TheraP [23], while in PSMAfore, nausea and anemia were increased ≥10% with [177]Lu-PSMA-617 vs. ARPI change (Figure 3) [27].

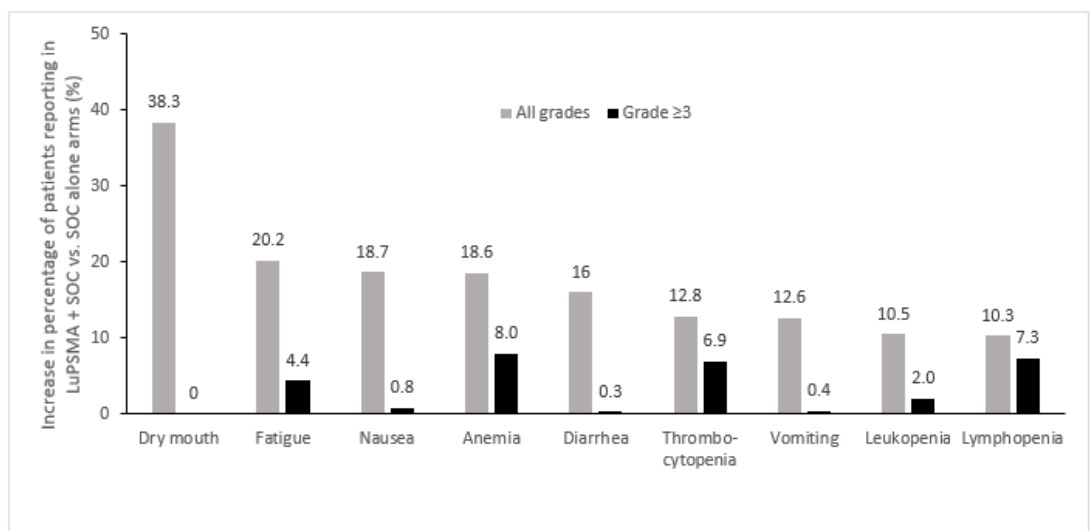

**Figure 2.** Adverse events most commonly increased in patients treated with [177]Lu-PSMA-617 + SOC vs. SOC alone in the phase 3 VISION trial (Δ ≥ 10%) [24]. LuPSMA, [177]Lu-PSMA-617; SOC, standard of care.

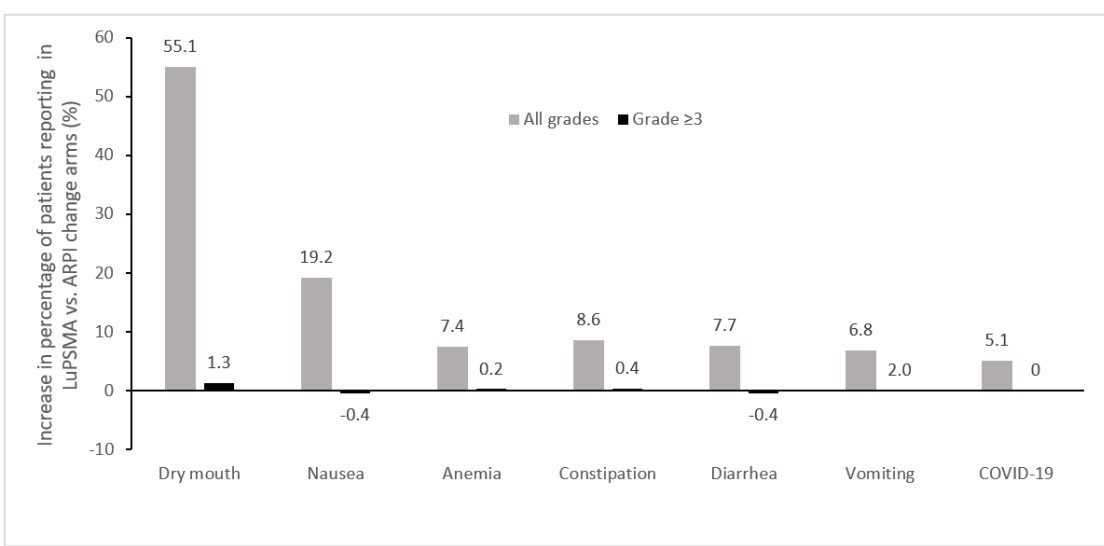

**Figure 3.** Adverse events most commonly increased in patients treated with [177]Lu-PSMA-617 vs. ARPI change in the phase 3 PSMAfore trial ($\Delta \geq 5\%$) [27]. ARPI, androgen receptor pathway inhibitor; LuPSMA, [177]Lu-PSMA-617.

## 4. Discussion

Clinical trials have demonstrated that [177]Lu-PSMA-617 improves imaging-based PFS and OS in patients with progressive, PSMA PET-positive mCRPC who have been pretreated with ARPIs and taxanes [23,24]. Improved imaging-based PFS vs. ARPI change was also demonstrated earlier in the disease course in taxane-naïve patients [27]. With health regulatory approval of [177]Lu-PSMA-617, the challenge is now to incorporate these clinical trial data into clinical practice.

### 4.1. Treatment Sequencing and Patient Selection Criteria

Selection of appropriate patients is key to fully realizing the potential benefits of RLT. While the Health Canada indication for [177]Lu-PSMA-617 requires that patients have been previously treated with at least one ARPI and one taxane [12], the optimal place of this therapy in the treatment sequence for mCRPC has yet to be determined. The current regulatory requirement for prior taxane exposure is based on the VISION clinical trial, which demonstrated the benefits of [177]Lu-PSMA-617 over SOC in taxane-exposed patients, including those who had received docetaxel as well as those who had received both docetaxel and cabazitaxel [24]. The TheraP trial demonstrated the superiority of [177]Lu-PSMA-617 for PFS as well as tolerability, although these patients were more highly selected for PSMA positivity by more stringent PSMA-PET criteria than the criteria used in the VISION study [23]. In addition, the current regulatory requirement for prior chemotherapy treatment is problematic as it has been demonstrated in population-based studies that the majority of patients with mCRPC within Canada never receive taxane chemotherapy during their disease course due to comorbidities that are common in this patient population, which is generally of advanced age [40]. Moreover, while patients who had been treated with radium-223 within six months were excluded from the VISION trial, real-world data suggest that treatment with radium-223 is feasible both before and after [177]Lu-PSMA-617 [41,42]. Other sequencing issues that have yet to be clarified include the benefits of RLT vs. docetaxel and its place in chemotherapy-naïve patients. Ongoing studies, such as Canadian Clinical Trials Group (CCTG) PR.21, PSMAddition, and PSMAfore, should help to address the optimal sequencing of [177]Lu-PSMA-617 in the current treatment paradigms for advanced prostate cancer [11].

While the results of these trials are awaited, no single criterion should preclude a patient who has already received ARPI and taxane treatment from consideration for treatment with [177]Lu-PSMA-617. Nonetheless, the European Association of Nuclear Medicine

(EANM)/SNMMI guideline for the use of [177]Lu-PSMA RLT suggests several factors that should be considered relative contraindications to treatment, such as life expectancy of less than six months, ECOG PS of more than two, severe myelosuppression, acute infections, acute bone complications (e.g., fracture, spinal cord compression), risk of multiorgan failure, untreated acute urinary tract obstruction, unmanageable urinary incontinence, unmanageable psychiatric comorbidities, and other severe comorbidities [43]. These factors should be considered in conjunction with the patient's overall health and cancer history, including time since diagnosis and the extent and location of metastases. An online risk calculator developed based on real-world nomograms (https://uclahealth.org/nuc/nomograms, accessed on 1 March 2024) [38] may assist oncologists with the selection of patients who should be considered with nuclear medicine for [177]Lu-PSMA-617, following prior ARPI and taxane treatment.

The results of PET imaging are also critical for determining patient suitability for therapy with [177]Lu-PSMA-617. While studies such as TheraP used dual PSMA and FDG PET imaging to determine PSMA-positivity [23], the VISION trial used PSMA PET/CT imaging for the inclusion of patients with at least one PSMA-positive metastatic lesion and no PSMA-negative lesions (Figure 4) [24,44]. This may be a reasonable alternative given that when a single-centre study examined 89 patients referred for [177]Lu-PSMA-617 with FDG and PSMA PET within two weeks, only three patients had an FDG/PSMA mismatch not detected by the PSMA PET-only (VISION-like) analysis [45], although the prevalence of $\geq 1$ FDG-positive/PSMA-negative lesion in the TheraP trial was 28% [23]. The EANM/SNMMI guideline suggests that while simultaneous FDG PET may be useful in certain cases, it is not mandatory for all patients [43]. From a Canadian perspective, dual PET imaging in all patients is not practical for many hospital centres given the limited availability of PET scanners and associated infrastructure [29]. However, while dual PET imaging is not necessary, a separate diagnostic contrast CT scan remains important as liver disease maybe not be evident on non-contrast CT acquired as part of PET/CT scans.

Regardless of the imaging methods used, higher SUVs ($SUV_{mean} \geq 10$) may be a prognostic or predictive biomarker that helps identify patients with more favourable prognosis [36,37]. Conversely, caution should be used and alternative therapies considered if available and applicable in patients with rapidly progressing disease or progressive visceral disease. In addition to appropriate imaging to determine RLT eligibility, multidisciplinary evaluation and discussion are important to determining whether to proceed with RLT for a specific patient given the existing and emerging spectrum of systemic therapy options [28].

*4.2. Necessary Facilities for Treatment Procedures*

The administration of [177]Lu-PSMA-617 requires dedicated treatment facilities for the administration of unsealed radiation sources. In particular, dedicated radiopharmacy facilities and treatment rooms are necessary, as well as standard operating procedures for patient isolation immediately after infusion and the management of contaminated materials after treatment until any residual radioactivity has decayed to safe levels for disposal through usual hospital waste streams. Drug administration should be performed by qualified technical personnel in appropriately licensed facilities supervised by physicians with appropriate training in the administration of radiopharmaceuticals. In most cases, treatments will be administered within nuclear medicine departments under the supervision of nuclear medicine physicians. In some jurisdictions, radiopharmaceutical administration may fall under the purview of radiation oncologists as this speciality is also well positioned to oversee these treatments given their training in therapeutic radiotherapy and dosimetry with external beam radiation as well as sealed and unsealed brachytherapy materials. Guidance documents for the administration of RLT have been issued by professional organizations such as the EANM, SNMMI, and American Society for Radiation Oncology (ASTRO), among others [43,46].

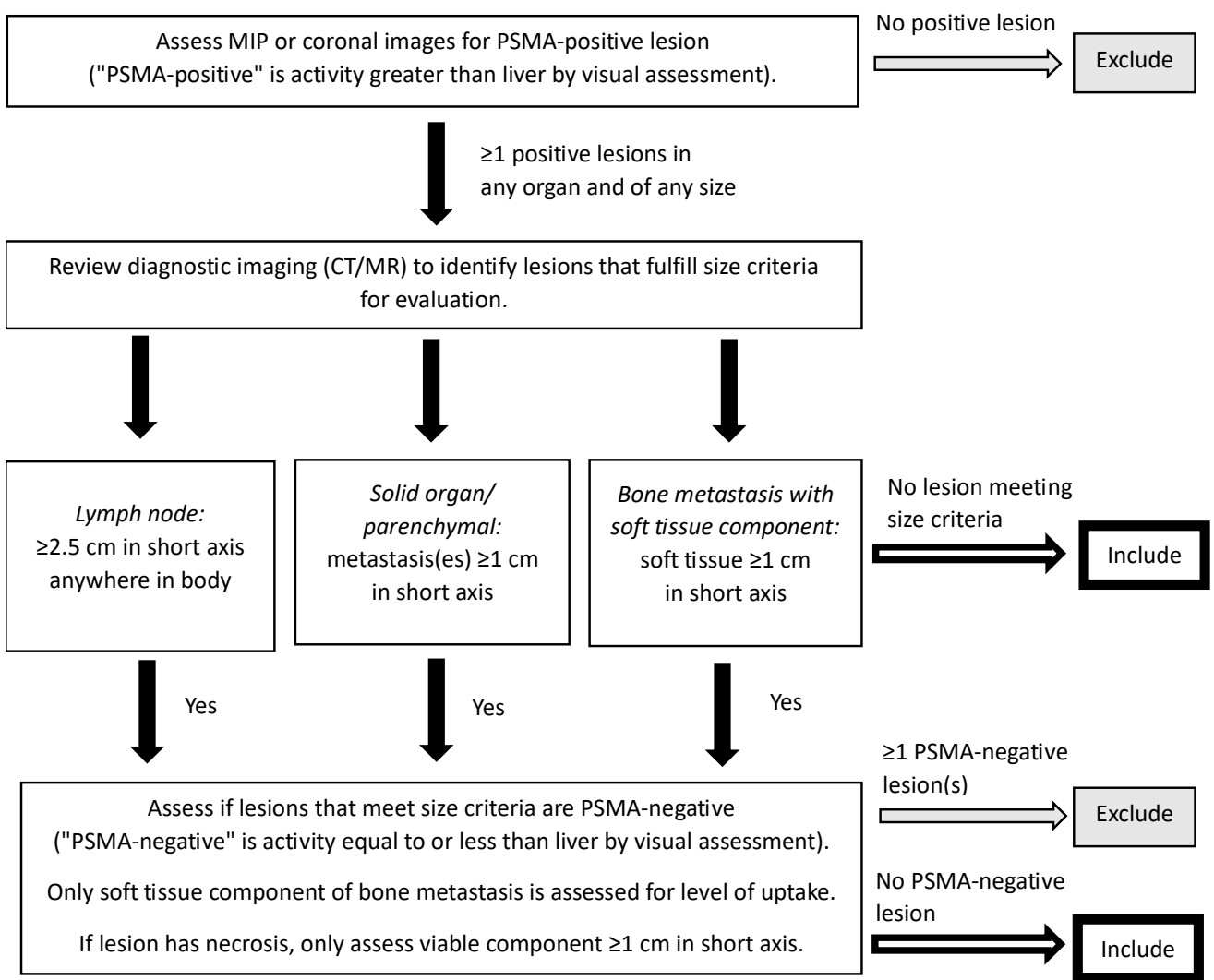

**Figure 4.** PSMA PET/CT selection criteria for the VISION trial [44]. A version of this figure was originally published in JNM. Kuo PH, Benson T, Messmann R, Groaning M. Why we did what we did: PSMA PET/CT selection criteria for the VISION trial. *J Nucl Med* **2022**, *63*, 816–818. © SNMMI [44]. CT, computed tomography; MIP, maximum intensity projection; MR, magnetic resonance; PET, positron emission tomography; PSMA, prostate-specific membrane antigen.

### 4.3. Counselling Patients on the Practicalities of Administration

Given the novelty of this therapeutic class in prostate cancer, patients who have been referred for $^{177}$Lu-PSMA-617 may have numerous questions regarding the real-world experience of RLT treatment. $^{177}$Lu-PSMA-617 is administered intravenously, often as an intravenous push given within one minute, with up to six doses being given at six-week intervals [12]. Typically, patients can expect to remain in the nuclear medicine department for 30–60 min. Although the radioactive nature of the therapy means certain precautions need to be taken in order to minimize radiation exposure to others, no hospitalization or prolonged isolation of patients is required due to radiation safety concerns and radioprotection may be managed by the patients at home, as outlined in sample patient instructions (Appendix A).

### 4.4. Assessment of Treatment Response

PSA should be monitored at each cycle in patients treated with $^{177}$Lu-PSMA-617 as PSA response becomes a reliable proxy for response 2–3 weeks after the second cycle [43]. In addition, the EANM/SNMMI guideline recommends that imaging-based

restaging be conducted every 12 weeks during treatment and at the end of each series of [177]Lu-PSMA RLT, with additional restaging conducted in cases of PSA rise (i.e., PSA increase of >25%) [43]. As in all mCRPC patients, the backbone of imaging restaging remains contrast CT scans of the chest, abdomen, and pelvis, and whole-body radionuclide bone scanning. Additionally, while restaging patients with PSMA PET or FDG PET was not carried out in the VISION trial and evidence supporting the post-therapeutic use of this strategy is limited, it may be useful in select patients where response or resistance would need to be determined in order to guide treatment decisions. Whole-body single photon emission computed tomography (SPECT) planar imaging, conducted 1–4 days post-therapy, may be another alternative to PSMA PET reimaging and response schemes based on SPECT imaging have been proposed [47,48]. In addition, the use of serial SPECT imaging for personalized [177]Lu-PSMA-617 dosimetry has been proposed to optimize treatment response [49]. However, at this time, such personalized dosimetry is not the standard of care as approvals for [177]Lu-PSMA-617 are for fixed dose administration, it is not yet possible to order and deliver patient-specific [177]Lu-PSMA-617 doses, and the benefits vs. fixed per patient dosing have yet to be established.

Therapy with [177]Lu-PSMA-617 may be continued until disease progression, unacceptable toxicity, or six cycles have been given [12]. However, there is no widely accepted definition as to what constitutes disease progression on PSMA imaging; for example, it is unclear whether the presence of one or two new lesions or, alternatively, increases in SUV in the absence of new lesions, would be considered progression. Additionally, it should be noted that if the imaging modality changes between staging and restaging, it can be difficult to differentiate true progression from pseudo-progression. In fact, in general, assessment of progression can be challenging in patients with mCRPC and discordant changes may be seen between PSA, imaging, and symptoms. Careful assessment and multidisciplinary review of cases is thus required to integrate all available information and make treatment decisions so as to not discontinue therapy too quickly in patients who may be benefiting.

### 4.5. Management of Adverse Events

As compared to SOC, the most common symptomatic adverse events seen with [177]Lu-PSMA-617 included dry mouth, fatigue, nausea, diarrhea, and cytopenias. Prophylactic antiemetic medication, such as ondansetron and/or corticosteroids, may help minimize nausea, and diarrhea may be managed through dietary changes and the use of medications such as loperamide or diphenoxylate/atropine [43]. Unfortunately, no effective strategies to manage treatment-related dry mouth or non-hematologic fatigue have yet been identified.

The treatment-modifying grade $\geq$ 3 adverse events most commonly observed with [177]Lu-PSMA-617 are hematologic in nature. Monitoring of hematologic parameters is thus advised before and during treatment with [177]Lu-PSMA-617 [12]. Treatment should be postponed or withheld in cases of grade $\geq$ 2 myelosuppression until recovery to baseline or grade 1 is observed [12,43]. Transfusion and/or erythropoietin may be used to manage anemia, while the use of growth factors may be appropriate for neutropenia [12,43].

It is important that RLT therapy be integrated into the multidisciplinary care of the patient with clear lines of communication and well-described roles and responsibilities regarding patient monitoring and treatment modification. For example, patients receiving RLT will be transitioning from prior ARPI and docetaxel chemotherapy and subsequently monitored by oncologists and nuclear medicine specialists. As patients move into a phase of treatment with RLT, it must be clear who is responsible for the monitoring and management of RLT toxicity; in most cases this will be the provider supervising RLT prescription and delivery. At the same time, co-management with the other specialties is required in order to ensure other oncologic issues, such as the administration of bone-modifying agents and intervention in cases of acute oncologic complications, are appropriately managed. Most patients will eventually experience treatment-limiting disease progression or toxicity and appropriate transition back to other oncologic specialties for alternate sys-

temic therapies and to palliative care specialists for supportive care treatments must occur efficiently and seamlessly.

*4.6. Ongoing Questions*

Despite the clear benefits seen in clinical trials of $^{177}$Lu-PSMA-617, a number of questions remain, including the identification of additional biomarkers to predict response, the implications of personalized dosimetry, the potential benefits of combination with other treatments, and the optimal sequencing with other therapies, including use earlier in the metastatic disease setting. Ongoing studies, such as CCTG PR.21, PSMAddition, and PSMAfore, among others, may help answer some of these questions [11]. Ongoing trials are also investigating the use of novel RLTs in progressive mCRPC, including treatments using $^{177}$Lu linked to different PSMA-binding or other prostate cancer-specific ligands as well as treatments using different radionuclides, such as actinium-225, iodine-131, and lead-212 [11]. The announcement of statistically significant topline results from the phase 3 SPLASH study of the PSMA-targeted RLT $^{177}$Lu-PNT2002 in patients with chemotherapy-naïve mCRPC who had progressed on an ARPI presages the advent of additional RLTs for patients with mCRPC [50].

**5. Conclusions**

$^{177}$Lu-PSMA-617 represents not just a new therapy but a new therapeutic class for the treatment of prostate cancer. As such, clinical pathways need to be developed and clinicians involved in the treatment of mCRPC must become familiar with these new processes in order to realize the benefits of RLT for their patients. While not all the suggestions included in this discussion are strictly evidence-based, it is the hope of the authors that this review of the evidence and associated expert opinions help practitioners translate these data into the current Canadian practice setting. Finally, as the therapeutic landscape for mCRPC continues to evolve, new treatments and emerging data will need to be considered when making treatment decisions.

**Author Contributions:** Conceptualization, G.B., K.N.C., U.E., S.J.H., C.K.K., D.T.L., T.N., F.P., R.R., F.S. and S.M.Y.; methodology, G.B., K.N.C., U.E., S.J.H., T.N., F.S. and S.M.Y.; validation, S.J.H. and S.P.; formal analysis, U.E., S.J.H., F.S. and S.M.Y.; investigation, F.S. and S.M.Y.; data curation, C.K.K. and S.M.Y.; writing—original draft preparation, K.N.C., U.E., F.P., S.P., F.S. and S.M.Y.; writing—review and editing, G.B., K.N.C., U.E., S.J.H., C.K.K., D.T.L., P.M., T.N., F.P., S.P., R.R., F.S. and S.M.Y.; visualization, T.N.; supervision, C.K.K., S.P., F.S. and S.M.Y.; project administration, C.K.K. All authors have read and agreed to the published version of the manuscript.

**Funding:** Financial support for medical writing assistance was provided by Novartis Pharmaceuticals Ltd. Canada. The authors had full control of the content and made the final decision for all aspects of this article.

**Acknowledgments:** We thank Rebecca Cowan for their medical editorial assistance with this manuscript.

**Conflicts of Interest:** G.B. has received compensation from Advanced Accelerator Applications/ Novartis and the Ontario Institute for Cancer Research. K.N.C. has served as a consultant and received honoraria from Advanced Accelerator Applications/Novartis, Astellas Pharma, AstraZeneca, Genentech/Roche, Janssen, Merck, Pfizer, and POINT Biopharma, and received grants and research funding from Advanced Accelerator Applications/Novartis, AstraZeneca, Genentech/Roche, Janssen, Pfizer, and POINT Biopharma. U.E. has received personal fees for serving as an advisor/consultant for Advanced Accelerator Applications/Novartis, Amgen, Astellas Pharma, AstraZeneca, Bayer, Ferring Pharmaceuticals, Janssen, Knight Therapeutics, Merck, and Pfizer, research funding from Astellas Pharma, Bayer, and Janssen, and institutional funding for the conduct of clinical studies from Advanced Accelerator Applications/Novartis, Astellas Pharma, AstraZeneca, Bayer, Genentech/Roche, Janssen, Merck, and POINT Biopharma. S.J.H. has received honoraria for advisory activities and has conducted research with funding paid to the institution from Advanced Accelerator Applications/Novartis. C.K.K. has received honoraria for presentations from Astellas Pharma, Bayer, Bristol-Myers Squibb, Eisai, Ipsen, Janssen, Merck, Pfizer, and Seagen, and for ad-hoc advi-

sory boards from Advanced Accelerator Applications/Novartis, Astellas Pharma, Bayer, BioNTech, Bristol-Myers Squibb, Eisai, Ipsen, Janssen, Merck, Pfizer, Seagen, and Sanofi. D.T.L. has participated in advisory meetings for Advanced Accelerator Applications/Novartis, Bayer, and Ipsen, has participated in research studies sponsored by Advanced Accelerator Applications/Novartis, POINT Biopharma, and Progenics, and has received consultancy fees from Advanced Accelerator Applications/Novartis and Ipsen. P.M. has served as a consultant and received honoraria from Advanced Accelerator Applications/Novartis. T.N. has received research/educational funding from AbbVie, Amgen, Astellas Pharma, AstraZeneca, Bayer, Janssen, Sanofi, TerSera, and Tolmar, and honoraria from AbbVie, Advanced Accelerator Applications/Novartis, Amgen, Astellas Pharma, AstraZeneca, Bayer, Ferring Pharmaceuticals, Janssen, Knight Therapeutics, Merck, Sanofi, TerSera, and Tolmar. F.P. has received honoraria and/or was a consultant for AbbVie, Advanced Accelerator Applications/Novartis, Amgen, Astellas Pharma, AstraZeneca, Bayer, Janssen, Merck, Sanofi, TerSera, and Tolmar. S.P has received speaking and consulting fees from Advanced Accelerator Applications/Novartis, Bayer, and POINT Biopharma. R.R. has served on advisory boards and speaker bureaus for AbbVie, Amgen, Astellas Pharma, AstraZeneca, Bayer, Ferring Pharmaceuticals, Janssen, McKesson, Sanofi, and TerSera, and has participated in clinical trials funded by Advanced Accelerator Applications/Novartis, Astellas Pharma, AstraZeneca, Bayer, Ferring Pharmaceuticals, Janssen, Myovant Sciences, Pfizer, POINT Biopharma, and Sanofi. F.S. has acted in a consulting/advisory role for and received honoraria from AbbVie, Advanced Accelerator Applications/Novartis, AstraZeneca, Astellas Pharma, Bayer, Janssen, Merck, Pfizer, and Tolmar, and has conducted research with funding paid to the institution from Advanced Accelerator Applications/Novartis, AstraZeneca, Astellas Pharma, Bayer, Janssen, Merck, and Pfizer. S.M.Y. has received consultant, advisory board, or sponsorship honoraria from Advanced Accelerator Applications/Novartis, AstraZeneca, Bayer, Bristol-Myers Squibb, Genentech/Roche, Ipsen, Janssen, Merck, OncoHelix, and Pfizer. Financial support for medical writing assistance was provided by Novartis Pharmaceuticals Ltd. Canada. The funder had no role in the design of the study, in the collection, analyses, or interpretation of the data, in the writing of the manuscript, or in the decision to publish the results.

## Appendix A. Literature Search Strategy

- Objective: to identify clinical trials and observational studies on the use of $^{177}$Lu-PSMA-617 in patients with prostate cancer that were published in the literature within the last 10 years or presented at a major congress within the last 3 years.
- Searches conducted:
  - Google Scholar
    - Search string: allintitle: (177Lu OR "lutetium-177" OR Lu OR lutetium) AND (PSMA OR "PSMA-617" OR "prostate specific membrane antigen" OR "vipivotide tetraxetan")
    - Limits: 2013 or more recent; terms in title
  - PubMed
    - Search string: ("prostate cancer" or "Prostatic Neoplasms" [Mesh]) AND (177Lu OR lutetium-177 OR Lu OR lutetium OR "Lutetium" [Mesh]) AND (PSMA OR PSMA-617 OR "prostate-specific membrane antigen" OR "vipivotide tetraxetan")
    - Limits: English language; article types: case reports, clinical study, clinical trial, comparative study, meta-analysis, observational study, randomized controlled trial; 2013 or more recent
  - ASCO database:
    - Search strings:
      - i. 177Lu
      - ii. Lutetium
    - Limits: ASCO and ASCO-GU conferences; years 2020, 2021, 2022, and 2023; topic: prostate cancer
  - ESMO database:

- ■ Search strings:
  i. 177Lu
  ii. Lutetium
- ■ Limits: meeting resources; tumour site: prostate cancer; years 2020, 2021, 2022, and 2023
  ○ SNMMI congress abstract supplements:
  - ■ Search string: 177Lu OR lutetium
  - ■ Years searched: 2020, 2021, 2022, and 2023.

## Appendix A. 1777Lu-PSMA Therapy Instructions for Patients

Your doctors have determined that $^{177}$Lu-PSMA therapy is the best way to treat your prostate cancer. Although safe, we need your help to minimize radiation exposure to the general population and members of your family following your therapy.

*Instructions*

1. Preferably, drive home alone after your treatment. If this is not possible, keep as much distance as possible between yourself and the driver.
2. To minimize radiation exposure to other people, keep a maximum distance and a minimum exposure time between yourself and anyone else. Spend the least amount of time necessary in close contact (stay more than 2 m away) with other people for the next 3 days. For example, sleep alone for the first 3 nights.
3. Avoid all contact with children less than 10 years of age for 7 days and with pregnant women for 15 days.
4. You can return to daily activities or work as early as 3 days after treatment, while avoiding contact with pregnant women and children less than 10 years of age.
5. Drink lots of water after the treatment and for the next 24 h (eight 8-ounce glasses).
6. Always follow good hygiene practices. Take at least one shower per day. You must use toilet paper each time you urinate. Wash your hands thoroughly after using the toilet. You should sit while urinating to avoid splashing. Flush the toilet twice after each use for the first 24 h. Caregivers must wear disposable gloves for 3 days after treatment if there is a risk of contact with bodily fluids.
7. If you have any nausea or vomiting, take the medication prescribed to you.
8. If you are planning to travel outside of the country by any means or to go to an airport in the next 3 months, please inform the Nuclear Medicine Department and you will be provided with a document explaining the therapy you just received.
9. Keep this document on you for the next week, and show it to your health care provider(s) should you require any urgent care in the next 7 days. Outside of working hours, health care providers can contact a nuclear medicine physician at TELEPHONE NUMBER.
10. Should you have questions regarding your treatment, you can contact someone during working hours at the Department of Nuclear Medicine at TELEPHONE NUMBER.

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
