# Peer review of "177Lu-PSMA-617 in Metastatic Castration-Resistant Prostate Cancer: A Review of the Evidence and Implications for Canadian Clinical Practice"

_curroncol, doi:10.3390/curroncol31030106_

Round 1

Reviewer 1 Report

Comments and Suggestions for Authors

The manuscript is well written but few points should be improved

1. The mergent role of PSMA PET/CT also in the initial diagnosis of prostate cancer (especially in men with high risk) should be added in the text (Pepe P, Pennisi M. Targeted Biopsy in Men High Risk for Prostate Cancer: 68Ga-PSMA PET/CT Versus mpMRI. Clin Genitourin Cancer. 2023 Dec;21(6):639-642)

2. The lower accuracy of PSMA PET/CT in the diagnosis, staging and follow up in men with cribiform and ductal prostate cancer should be added in the Discussion

Author Response

While the emerging role of PSMA PET/CT in the diagnosis, staging, and follow-up of prostate cancer is an important and evolving one (as mentioned in comments #1 and #2), it is somewhat outside the scope of the current review, which focuses specifically on the therapeutic use of the radioligand therapy 177Lu-PSMA-617. Thus, to maintain the focus of the article and to avoid superficial discussion of the important topic of PSMA PET/CT in diagnosis, no additions regarding this topic were made. 

Reviewer 2 Report

Comments and Suggestions for Authors

This review manuscript provides a analysis of the findings from current phase 2 and 3 trials on the use of 177Lu-PSMA-617 in patients with mCRPC. As the PSMA-targeting therapy, 177Lu-PSMA-617 demonstrates considerable promise. The review highlights urgent aspects that merit further discussion, contributing to the literature. However, several areas require enhancement to augment the manuscript's value and clarity:

1. The title suggests a broad focus on prostate cancer, which encompasses a wider spectrum than the specific mCRPC context discussed. More discussion on the feasibility and potential of 177Lu-PSMA-617 in treating other forms of PCa are needed.

2. The introduction would benefit from an expanded overview of PSMA-617, including its properties and mechanisms. A comparative analysis with other PSMA-targeting ligands could provide a clearer understanding of its unique advantages and positioning within the landscape of prostate cancer treatments.

3. Additional details about 177Lu, including its radioactive decay characteristics, half-life, and the energy of the emitted particles, would enrich the manuscript. Such information is crucial for understanding the therapeutic mechanism and safety profile of ^177Lu-PSMA-617.

4. An in-depth discussion on PSMA expression across different stages and types of prostate cancer, as well as other organs would be valuable.

5. Incorporating figures in sections 3.3 and 3.4 could significantly enhance the presentation of results and provide an intuitive understanding.

6. The readability of the manuscript could be improved by reorganizing the text in lines 158-165.

7. An analysis of how 177Lu-PSMA-617 impacts the overall cost of mCRPC treatment is absent.

Comments on the Quality of English Language

N/A

Author Response

To address comment #1, the title has been revised to reflect the focus of the article on metastatic castration-resistant prostate cancer. Additional details on the properties of both 177Lu and PSMA-617 have been added to the introduction (see lines 73-76) in response to comments #2 and #3. However, no comparative analysis was performed as the review did not encompass radioligand therapies that are not currently approved in the Canadian setting, and this was deemed out of the scope of this review. Additional details on PSMA expression and localization have also been added to the introduction (lines 63-65) to address comment #4. In response to comment #5, a new table (Table 2) summarizing survival and quality of life outcomes of randomized trials has been added. Lines 218 to 227 (previously 158-165) were revised for clarity to address comment #6. As regards comment #7, CADTH’s estimate of additional costs per patient has been added (see lines 92-96).

Reviewer 3 Report

Comments and Suggestions for Authors

It is an excellent review regarding the Lu-PSMA in the practice of prostatic cancer in Canada. I would like to congratulate the authors for the excellent review of the articles' databases and the very structured presentation of the outcomes.

The detailed discussion regarding the counseling to the patients receiving Lu-PSMAand the management of the adverse events are very beneficial.

Minor-Revision

The authors should present a PRISMA flowchart concerning the review of the literature.

Author Response

The information previously presented in “Appendix B. Results of literature review” has been moved to the main body of the text and reformatted as a PRISMA flowchart (now Figure 1). 

Round 2

Reviewer 2 Report

Comments and Suggestions for Authors

The revision addressed all my concerns.